

# Functional assessment of chronic illness therapy—the fatigue scale exhibits stronger associations with clinical parameters in chronic dialysis patients compared to other fatigue-assessing instruments

Chia-Ter Chao[1,2,3], Jenq-Wen Huang[3], Chih-Kang Chiang[2,4] and COGENT (COhort of GEriatric Nephrology in NTUH) study group

[1] Department of Medicine, National Taiwan University Hospital Jinshan branch, New Taipei City, Taiwan
[2] Graduate Institute of Toxicology, National Taiwan University College of Medicine, Taipei, Taiwan
[3] Department of Internal Medicine, National Taiwan University Hospital, Taipei, Taiwan
[4] Department of Integrative Diagnostics and Therapeutics, National Taiwan University Hospital, Taipei, Taiwan

Corresponding author
Chih-Kang Chiang,
ckchiang@ntu.edu.tw

## ABSTRACT

**Background.** Patients with end-stage renal disease (ESRD) have a high symptom burden, among which fatigue is highly prevalent. Many fatigue-assessing instruments exist, but comparisons among instruments in this patient population have yet to be investigated.

**Methods.** ESRD patients under chronic hemodialysis were prospectively enrolled and seven types of fatigue instruments were administered: Brief Fatigue Inventory (BFI), Functional Assessment of Chronic Illness Therapy–Fatigue (FACIT-F), Fatigue Severity Scale (FSS), Lee Fatigue Scale (LFS), Fatigue Questionnaire (FQ), Fatigue Symptom Inventory (FSI), and Short-Form 36-Vitality (SF36-V). Using these instruments, we investigated the correlation between fatigue severity and clinical/biochemical parameters, including demographic/comorbidity profile, dialysis-related complications, and frailty severity. We used regression analysis with serum albumin and frailty severity as the dependent variables to investigate the independent correlations.

**Results.** A total of 46 ESRD patients were enrolled (average age of $67 \pm 11.6$ years), and 50% of them had type 2 diabetes mellitus. Results from the seven tested instruments showed high correlation with each other. We found that the fatigue severity by FACIT-F was significantly associated with age ($p = 0.03$), serum albumin ($p = 0.003$) and creatinine ($p = 0.02$) levels, while SF36-V scores were also significantly associated with age ($p = 0.02$) and serum creatinine levels ($p = 0.04$). However, the fatigue severity measured by the FSS, FSI, FQ, BFI, and LFS did not exhibit these associations. Moreover, regression analysis showed that only FACIT-F scores were independently associated with serum albumin levels and frailty severity in ESRD patients.

**Conclusion.** Among the seven fatigue-assessing instruments, only the FACIT-F yielded results that demonstrated significant and independent associations with important outcome-related features in ESRD patients.

# INTRODUCTION

Patients with end-stage renal disease (ESRD) are living longer than ever due to the advancement of medical care, early diagnosis and the improvement in treating chronic kidney disease (CKD). However, this increased survival is accompanied by more physical discomfort. Patients with ESRD on chronic dialysis endure high symptom burden throughout the progression of CKD, and this condition persists even after dialysis commencement (*Thong et al., 2009*). The totality of symptom burden describes the subjective discomfort that negatively influences patients physically, psychologically, socially, and emotionally. The extensive symptomatology in this population carries clinical importance since it significantly lowers ESRD patients' health-related quality of life (HRQoL); in fact, some reports have suggested better outcome-predictive efficacy of self-rated health than traditional biomarkers (*Han et al., 2009*; *Robinson-Cohen et al., 2014*).

Among the spectrum of symptoms in ESRD patients, fatigue, a sense of weakness or lack of energy, assumes the highest prevalence; epidemiologic surveys yield that 49–92% of chronic dialysis patients reported fatigue during some time in their dialysis tenure (*Davison, Jhangri & Johnson, 2006*; *Weisbord et al., 2007*; *Son et al., 2009*). Fatigue has been found to be associated with biochemical features including serum albumin, hemoglobin, calcium, phosphate, glycemic status, C-reactive protein, and cytokines in different populations (*Karakan, Sezer & Ozdemir, 2011*; *Ormstad et al., 2011*; *Rat et al., 2012*; *Chilcot et al., 2015*). Fatigue among ESRD patients can result from complications arising from the loss of renal function *per se*, including renal anemia, the malnutrition-inflammation complex, and the accumulation of uremic toxins as well as from physical inactivity, psychological impairments, and administered treatments such as medications and dialysis-related issues.

Measuring fatigue is part of a systemic effort that attempts to address the entire spectrum of symptom burden in patients with different illnesses, especially cancer. Single-item–based assessments are a common practice (''Are you feeling tired?'' or ''Do you suffer from a constant lack of energy?''); however, in light of the high prevalence and clinical importance of fatigue, questionnaire-based tools have been devised to better characterize the severity and potential influences of fatigue using a format similar to that for measuring HRQoL (*Unruh, Weisbord & Kimmel, 2005*). The Short Form-36 Vitality subscale (SF36-V) might be the most widely used fatigue-measuring instrument for research purposes; however, the most valid instrument in ESRD patients remains controversial. Moreover, none of the existing studies have attempted to compare the clinical significance of different fatigue-assessing instruments in this patient population. Accordingly, the current study aimed to evaluate the efficacy of seven existing fatigue-assessing instruments for predicting clinically important parameters in ESRD patients.

## MATERIALS AND METHODS

### Ethical considerations

The current study was approved by the institutional review board of National Taiwan University Hospital (NTUH; NO. 201403006RINB), and all participants (or their medical proxies if they were unable to communicate verbally) provided verbal informed consent.

### Collection of baseline clinical parameters

Patients with ESRD receiving chronic hemodialysis at the NTUH Jinshan branch were prospectively recruited between 2014 and 2015 (*Chao et al., 2015*; *Chao & Huang, 2015*). Those who refused to provide informed consent were excluded from this study. We documented their baseline clinical data on enrollment, including demographic profile (age, sex, education status, and other socioeconomic factors), body mass index (BMI), dialysis duration, and ESRD origin. Comorbidity severity was recorded using Liu's dialysis comorbidity index (*Liu et al., 2009*). Blood samples were obtained after enrollment and sent to the reference laboratory of NTUH for analysis of hemoglobin, nutritional and biochemical profiles (albumin, cholesterol, triglyceride, uric acid, pre-dialysis blood urea nitrogen [BUN], creatinine, and electrolyte panels), dialysis clearance, and hormones (ferritin, intact parathyroid hormone) (*Chao et al., 2012*). Assuming a power of 0.8 and a type 1 error of 5%, at least 32 patients will be required to detect significant differences in serum albumin levels among ESRD patients with and without fatigue.

### Assessment of fatigue using different self-report instruments

Trained research assistants administered all of the fatigue assessment tools during the patients' dialysis sessions nearest to their time of blood collection. We administered these instruments in a random order. A total of seven instruments were used in this study, including the Brief Fatigue Inventory (BFI), Functional Assessment of Chronic Illness Therapy–Fatigue (FACIT-F), Fatigue Severity Scale (FSS), Lee Fatigue Scale (LFS), Fatigue Questionnaire (FQ), Fatigue Symptom Inventory (FSI), and SF36-V. Six out of the seven tools (BFI, FACIT-F, FSS, LFS, FSI, and SF36-V) have been translated into Chinese versions in the literature with established consistency and reliability (*Wang et al., 2004*; *Shun et al., 2006*; *Tsai et al., 2014*; *Wang et al., 2016*). For those without a Chinese version, at least two staff members performed a forward and backward translation, and any discordant aspects were resolved by group consensus.

Fatigue assessment tools can be stratified according to the dimensions they intend to cover (uni- or multidimensional) (*Minton & Stone, 2009*). Here we evaluated the utility of both uni- (BFI, FACIT-F, FSS) and multidimensional (LFS, FQ, FSI, SF36-V) instruments for chronic dialysis patients. The BFI, a nine-item questionnaire that was originally developed to evaluate fatigue severity in cancer patients, features the advantages of brevity and avoidance of English-based idioms that tend to confuse Chinese recipients (*Mendoza et al., 1999*). Scores can be calculated as the average of all the items assessed (range, 0–10). The FACIT-F, part of a comprehensive assessment tool that evaluates HRQoL and the symptomatology of patients with chronic illnesses or cancer, consists of 13 items graded on a four-point Likert scale (http://www.facit.org). The FACIT-F has been used in chronic

dialysis patients with good validity (content validity index 0.67–1; good construct validity from an excellent correlation with revised Piper Fatigue Scale results) (*Wang et al., 2015*). The FSS, a nine-item test graded using a seven-point Likert scale, was initially tested in patients with neurological disorders but has since been expanded for use in patients with other diseases and aims to measure fatigue severity and its influence on one's daily activity and lifestyle (*Krupp et al., 1989*). The original LFS consists of 18 items graded on a 10-point Likert scale and is used to assess patient fatigue or energy with different descriptive adjectives (*Lee, Hicks & Nino-Murcia, 1991*). We previously used a seven-item short-form LFS to increase administration ease along with similarly fair psychometric properties (*Tsai et al., 2014*). LFS scores are calculated similarly to BFI scores. FQ, or the original Chalder fatigue scale, evaluates two dimensions of patient fatigue, physical and mental, and features the advantage of cultural sensitivity (*Chalder et al., 1993*). The FSI, which also uses a 10-point Likert scale, measures fatigue intensity at different timings (severity dimension), how fatigue affects patients' functional statuses (interference dimension), and the persistence of fatigue over time (duration dimension) (*Hann et al., 1998*). The SF36-V is an integral component of the SF36, which systematically surveys the quality of life of patients with diverse disease forms. The SF36-V is now widely used to gauge fatigue severity and energy levels of ill patients, including those with ESRD (*Feroze et al., 2011*).

## Frailty assessment using an established instrument

Frailty was assessed in these ESRD patients using the Simple FRAIL Scale (SFS). The SFS measures five domains of frailty: fatigue, resistance, ambulation, illnesses, and loss of body weight. The SFS has been applied in ESRD patients with fair validity, the results of which have shown moderate associations with dialysis-related complications (*Chao et al., 2014a*; *Chao et al., 2014b*).

## Statistical analysis

In this study, continuous and categorical variables were described using mean ± standard deviation and case number with percentage, with group comparisons made using the independent $t$-test or the Mann–Whitney $U$-test and chi-square test, respectively. Since diabetes mellitus (DM) and heart failure (HF) might have an influence on fatigue, we also compared the severity of fatigue between ESRD patients with and without DM or HF. Results from different fatigue questionnaires were tested with respect to their association with the clinical parameters (demographic and socioeconomic features), dialysis-related complications (nutritional profiles, azotemia, electrolyte panels, and metabolic results), and frailty severity using Pearson's correlation coefficients. We then performed multiple regression analyses, incorporating relevant clinical features with dependent variables including serum albumin levels and frailty severity, both important outcome-related factors of ESRD patients. In all analyses, a two-sided $p < 0.05$ signified statistical significance as determined by SPSS 18.0 software (SPSS Inc., Chicago, IL, USA).

## RESULTS

### Features of the enrolled chronic dialysis patients

A total of 46 ESRD patients were recruited and participated in the study, higher than the calculated required sample size. Their average age was $67 \pm 11.6$ years, comparatively higher than that of an average dialysis population (*Chao et al., 2014a*; *Chao et al., 2014b*), although the male/female distribution was even (Table 1). Most patients were married, and one-third of our total cohort had no educational history, while another one-third went to elementary school only. The mean patient BMI was $23.2 \pm 3.1$ kg/m$^2$. Diabetes nephropathy was the cause of the ESRD in half of our patients. The dialysis comorbidity index of our patient cohort was $1.54 \pm 1.76$.

The serum biochemical profiles of these patients indicated that they had fair nutrition status (albumin, $3.9 \pm 0.4$ g/dL; total cholesterol, $160 \pm 45$ mg/dL), mild anemia (hemoglobin $10.2 \pm 3.6$ mg/dL), fair dialytic clearance, and nearly normal electrolyte panels (Table 1).

Among our cohort, half were found to have DM. No significant differences were found between ESRD patients with and without DM regarding demographic profiles, body mass index, and comorbidities including heart failure. Similarly, there were no significant differences regarding their nutritional profiles, azotemic parameters, electrolyte panels, and metabolic profiles) (Table 1). However, ESRD patients with DM had a significantly higher dialysis comorbidity index ($p = 0.01$) and were more likely to have ESRD originating from diabetic nephropathy ($p = 0.001$), both of which were expectable judging from the influence of DM. On the other hand, no significant differences were found between ESRD patients with and without HF regarding demographic profiles, body mass index, and all the laboratory data tested.

### Fatigue severity among chronic dialysis patients

In our study, FSI and FQ were translated by our staffs, while other questionnaires were translated previously. All patients answered the seven self-report questionnaires, and the fatigue assessment results are shown in Table 2. Using the previously reported diagnostic threshold for each questionnaire (for FACIT-F, score < 30; for FSI, score > 3; for FSS, score > 2.3; for FQ, score > 15; for SF36-V, score < 60; for LFS and BFI, score > 0), we found that the BFI identified fatigue in 65.2% of the patients, while the LFS identified fatigue in 56.6% of the patients. However, the FQ, FSS, and SF36-V identified fatigue in 21.7% of the patients each, with average scores of $14.6 \pm 4.1$, $1.8 \pm 1.2$, and $77 \pm 15$, respectively. On the other hand, the FACIT-F identified fatigue in 15.2% of the patients, with a mean score of $41.1 \pm 9.9$. According to the dimensions each questionnaire cover, no significant differences were observed between physical and mental components (using the FQ), while energy subscale was more likely to be affected than fatigue subscale (using the SF36-V).

In addition, the presence of DM did not have significant influences on these patients' fatigue severity, assessed by LFS ($p = 0.61$), BFI ($p = 0.84$), FQ ($p = 0.89$), FACIT-F ($p = 0.56$), FSI ($p = 0.83$), FSS ($p = 0.88$), and SF36-V ($p = 0.67$). Similarly, the presence of HF did not significantly affect fatigue severity, assessed by LFS ($p = 0.61$), BFI ($p = 0.56$), FQ ($p = 0.49$), FACIT-F ($p = 0.57$), FSI ($p = 0.41$), FSS ($p = 0.34$), and SF36-V ($p = 0.37$).

**Table 1  Baseline characteristics of enrollees.**

| Clinical features | DM | Non-DM | p value |
|---|---|---|---|
| **Demographic profile** | | | |
| Age (years) | 69.4 ± 11.7 | 66 ± 11.6 | 0.3 |
| Gender (male %) | 11 (48) | 11 (48) | 1 |
| Dialysis duration (years) | 2.6 ± 2.3 | 4.1 ± 3.2 | 0.07 |
| Marriage (yes %) | 22 (96) | 21 (91) | 0.89 |
| *Education (%)* | | | 0.9 |
| None | 8 (35) | 9 (39) | |
| Elementary school | 10 (43) | 8 (35) | |
| High school or higher | 5 (22) | 6 (26) | |
| Body mass index (kg/m$^2$) | 23.5 ± 2.5 | 22.9 ± 3.5 | 0.51 |
| **Comorbidity** | | | |
| Heart failure (%) | 5 (22) | 4 (17) | 0.84 |
| Dialysis comorbidity Index | 2.24 ± 1.64 | 0.96 ± 1.65 | 0.01 |
| **ESRD origin (%)** | | | 0.001 |
| Diabetic nephropathy | 19 (83) | 0 (0) | |
| Chronic glomerulonephritis | 2 (9) | 2 (9) | |
| Others | 1 (4) | 6 (26) | |
| Unknown | 1 (4) | 15 (65) | |
| **Laboratory data** | | | |
| *Nutritional profiles* | | | |
| Albumin (g/dL) | 3.9 ± 0.4 | 4 ± 0.4 | 0.36 |
| Total cholesterol (mg/dL) | 158 ± 44 | 161 ± 47 | 0.8 |
| Triglyceride (mg/dL) | 186 ± 120 | 145 ± 125 | 0.26 |
| HDL (mg/dL) | 40 ± 9.1 | 43 ± 13 | 0.29 |
| LDL (mg/dL) | 86 ± 37 | 93 ± 33 | 0.53 |
| Uric acid (mg/dL) | 8.5 ± 2.2 | 8.6 ± 1.3 | 0.84 |
| *Azotemic parameters* | | | |
| BUN (mg/dL) | 87.4 ± 22.7 | 86.8 ± 19.1 | 0.92 |
| Cre (mg/dL) | 11 ± 2.4 | 11.6 ± 2.3 | 0.35 |
| Kt/V | 1.6 ± 0.3 | 1.6 ± 0.2 | 0.84 |
| Urea reduction ratio (%) | 74.3 ± 5 | 74.7 ± 3.4 | 0.8 |
| *Electrolyte panels* | | | |
| Sodium (meq/L) | 134 ± 3.2 | 135 ± 3.8 | 0.31 |
| Pottasium (meq/L) | 4.7 ± 0.7 | 4.8 ± 0.7 | 0.4 |
| Caclium (mg/dL) | 9.1 ± 0.9 | 9.3 ± 0.6 | 0.29 |
| Phosphrus (mg/dL) | 5.3 ± 1.6 | 4.4 ± 1 | 0.02 |
| *Metabolic/hormones* | | | |
| Hemoglobin (mg/dL) | 10.2 ± 1.7 | 12.2 ± 11.7 | 0.44 |
| Ferritin (ng/mL) | 1191 ± 2048 | 597 ± 293 | 0.18 |
| TSAT (%) | 30.9 ± 23.6 | 26.6 ± 10.7 | 0.43 |
| Intact PTH (pg/ml) | 361 ± 374 | 299 ± 254 | 0.52 |

**Notes.**

BUN, blood urea nitrogen; DM, diabetes mellitus; ESRD, end-stage renal disease; HDL, high density lipoprotein; LDL, low density lipoprotein; PTH, parathyroid hormone; TSAT, transferrin saturation.
**Table 2  Results of fatigue instrument assessment in the current cohort (score range in parentheses).**

| Questionnaire | Survey results and dimensions explored | | |
|---|---|---|---|
| **Uni-dimensional** | **Total** | **DM** | **Non-DM** |
| Brief fatigue inventory | 1.5 ± 2.1 (0–8.7) | 1.5 ± 1.9 (0–7.2) | 1.6 ± 2.4 (0–8.7) |
| FACIT-fatigue | 41.1 ± 9.9 (13–52) | 40.3 ± 10.6 (13–52) | 42 ± 9.3 (21–52) |
| Fatigue severity scale | 1.8 ± 1.2 (0.9–5.8) | 1.8 ± 1.1 (1.1–5.4) | 1.7 ± 1.3 (0.9–5.8) |

**Multi-dimensional**

| | Composite | | | Fatigue | | | Energy | | |
|---|---|---|---|---|---|---|---|---|---|
| | **Total** | **DM** | **Non-DM** | **Total** | **DM** | **Non-DM** | **Total** | **DM** | **Non-DM** |
| Lee fatigue scale | 1.3 ± 1.8 (0–6.6) | 1.4 ± 1.9 (0–6.6) | 1.1 ± 1.7 (0–5.4) | 1.3 ± 1.9 (0–7.7) | 1.4 ± 2 (0–7.7) | 1.1 ± 1.9 (0–6.9) | 1.3 ± 2.1 (0–10) | 1.5 ± 2.4 (0–10) | 1.1 ± 1.9 (0–7.3) |

| | Composite | | | Physical | | | Mental | | |
|---|---|---|---|---|---|---|---|---|---|
| | **Total** | **DM** | **Non-DM** | **Total** | **DM** | **Non-DM** | **Total** | **DM** | **Non-DM** |
| Fatigue question-naire | 14.6 ± 4.1 (0–25) | 14.7 ± 3.9 (3–25) | 14.5 ± 4.3 (0–24) | 8.5 ± 2.8 (0–16) | 8.6 ± 3 (1–16) | 8.4 ± 2.8 (0–15) | 6.1 ± 1.5 (0–10) | 6.1 ± 1.2 (2–10) | 6 ± 1.7 (0–10) |

| | Composite | | | Severity | | | Interference | | | Duration | | |
|---|---|---|---|---|---|---|---|---|---|---|---|---|
| | **Total** | **DM** | **Non-DM** | **Total** | **DM** | **Non-DM** | **Total** | **DM** | **Non-DM** | **Total** | **DM** | **Non-DM** |
| Fatigue symptom inventory | 1.3 ± 1.7 (0–7.2) | 1.3 ± 1.5 (0–5.5) | 1.4 ± 1.9 (0–7.2) | 1.5 ± 1.8 (0–6.3) | 1.6 ± 1.8 (0–6.3) | 1.4 ± 1.7 (0–5.8) | 1.1 ± 2.1 (0–8.4) | 1 ± 1.6 (0–5.9) | 1.3 ± 2.5 (0–8.4) | 1.8 ± 2 (0–6) | 1.7 ± 2 (0–6) | 1.8 ± 2.1 (0–6) |

| | Composite | | | Energy | | | Fatigue | | |
|---|---|---|---|---|---|---|---|---|---|
| | **Total** | **DM** | **Non-DM** | **Total** | **DM** | **Non-DM** | **Total** | **DM** | **Non-DM** |
| Short form 36-vitality | 77 ± 15 (44–100) | 76 ± 14.9 (44–96) | 77.9 ± 15.3 (44–100) | 33.5 ± 8.1 (18–50) | 32.8 ± 8 (18–46) | 34.2 ± 8.3 (18–50) | 43.5 ± 7.6 (26–50) | 43.2 ± 7.7 (26–50) | 43.7 ± 7.7 (26–50) |

**Notes.**

DM, diabetes mellitus; FACIT, functional assessment of chonic illness therapy.

**Table 3 Correlations between different fatigue questionnaire results.**

| Correlation Coefficient[*] | BFI | FACIT-F | FSS | LFS | FQ | FSI | SF36-V |
|---|---|---|---|---|---|---|---|
| BFI | | −0.71 | 0.95 | 0.84 | 0.47 | 0.98 | −0.74 |
| FACIT-F | −0.71 | | −0.67 | −0.77 | −0.43 | −0.77 | 0.92 |
| FSS | 0.95 | −0.67 | | 0.9 | 0.56 | 0.93 | −0.7 |
| LFS | 0.84 | −0.77 | 0.9 | | 0.57 | 0.87 | −0.79 |
| FQ | 0.47 | −0.43 | 0.56 | 0.57 | | 0.49 | −0.55 |
| FSI | 0.98 | −0.77 | 0.93 | 0.87 | 0.49 | | −0.81 |
| SF36-V | −0.74 | 0.92 | −0.7 | −0.79 | −0.55 | −0.81 | |

**Notes.**

*$p < 0.01$ for all the correlation analyses, by Pearson's correlation analyses.

BFI, brief fatigue inventory; FACIT-F, functional assessment of chronic illness therapy—fatigue; FQ, fatigue questionnaire; FSI, fatigue symptom inventory; FSS, fatigue severity scale; LFS, Lee fatigue scale; SF36-V, short-form 36-vitality subscale.

## Correlation between fatigue severity, clinical features, and frailty severity among chronic dialysis patients

A correlation analysis was performed of the assessment results and the clinical and laboratory parameters. Results from all seven fatigue-assessment tools showed high correlations (Table 3). We further found that the FACIT-F scores exhibited significant associations with age ($r = -0.33, p = 0.03$) as well as serum albumin ($r = 0.43, p = 0.003$), BUN ($r = 0.31, p = 0.04$), creatinine ($r = 0.35, p = 0.02$), and serum potassium ($r = 0.32, p = 0.03$) levels among all ESRD patients. SF36-V scores were also significantly associated with age ($r = -0.34, p = 0.02$) as well as BUN ($r = 0.29, p = 0.05$), creatinine ($r = 0.3, p = 0.04$), and serum potassium ($r = 0.3, p = 0.04$) levels but borderline associated with serum albumin ($r = 0.28, p = 0.07$) level among all ESRD patients. On the contrary, fatigue severity measured by the FSS ($r = 0.31, p = 0.04$), FSI ($r = 0.31, p = 0.04$), and BFI ($r = 0.36, p = 0.02$) was significantly associated with dialysis duration but not biochemical traits. In addition, the FACIT-F, SF36-V, and LFI scores correlated significantly with frailty severity (FACIT-F vs. SFS, $r = -0.52, p = 0.0002$; SF36-V vs. SFS, $r = -0.35, p = 0.02$; LFI vs. SFS, $r = 0.32, p = 0.03$), while BFI ($r = 0.24, p = 0.12$), FQ ($r = 0.2, p = 0.19$), FSI ($r = 0.25, p = 0.09$), or FSS ($r = 0.24, p = 0.11$) did not.

## Regression analysis targeting serum albumin levels incorporating fatigue severity

Since serum albumin is an important prognosis determinant for chronic dialysis patients, we next performed a multiple regression analysis with serum albumin as the dependent variable among all ESRD patients (Table 4). Accounting for age, sex, dialysis duration, BMI, and comorbidity index, lower FACIT-F scores and SF36-V scores were both independently associated with lower serum albumin (for the former, $p = 0.002$; for the later, $p = 0.03$). However, only the FACIT-F score retained its independent association with serum albumin ($p = 0.006$). Such an association was not discovered for the BFI ($p = 0.13$), FQ ($p = 0.99$), LFI ($p = 0.14$), FSS ($p = 0.29$), and FSI ($p = 0.12$) scores. Similarly, only the FACIT-F scores ($p = 0.006$) exhibited a significant association with the SFS scores in these patients, while the other scale scores did not.

**Table 4**  Regression analysis using the entire ESRD cohort, with serum albumin level as the dependent variable.

| Results | $\beta$ coefficient | $t$ value | $p$ value |
|---|---|---|---|
| **Model 1a** | | | |
| FACIT-F scores | 0.45 | 3.31 | 0.002 |
| **Model 1b** | | | |
| SF36-V scores | 0.33 | 2.29 | 0.03 |
| **Model 2** | | | |
| FACIT-F scores | 1.13 | 3.4 | 0.006 |

**Notes.**

Model components: adjusted for age, gender, dialysis duration, and dialysis comorbidity index.
Model 1a: with FACIT-Fatigue score only.
Model 1b: with SF36 vitality score only.
Model 2: with both FACIT-Fatigue score and SF36 vitality score.
ESRD, end-stage renal disease; FACIT-F, functional assessment of chronic illness therapy–fatigue; SF36-V, short-form 36-vitality subscale.

We further used the sub-cohort with DM for the regression analysis, incorporating the original model components (age, sex, dialysis duration, and BMI). We found that FACIT-F scores still exhibited significant association with serum albumin levels among ESRD patients with DM ($t = 2.38, p = 0.03$). However, SF36 scores did not have significant association with serum albumin levels among ESRD patients with DM ($t = 1.98, p = 0.06$). If SF36 and FACIT-F scores were both included in the analysis, only FACIT-F scores retained a significant relationship with serum albumin levels ($t = 2.38, p = 0.03$). On the other hand, if we used the sub-cohort without DM for regression analysis, no significant association was found between FACIT-F score and serum albumin, or between SF36 scores and serum albumin (for FACIT-F, $t = 1.88, p = 0.08$; for SF36, $t = 0.91, p = 0.38$). Using the non-HF sub-cohort, we found that both FACIT-F ($t = 3.22, p = 0.003$) and SF36 ($t = 2.35, p = 0.02$) scores exhibited significant associations with serum albumin levels. However, for the HF sub-cohort, the number was too small to permit regression analysis. These findings were largely compatible with results using the entire ESRD cohort.

## DISCUSSION

We administered seven different fatigue questionnaires to a cohort of chronic dialysis patients, and fatigue was found in 15–65% of these patients. All of the questionnaires yielded results with high correlations. We also discovered that the FACIT-F and SF36-V scale findings exhibited a significant association with patient age and their nutritional parameters (serum albumin, creatinine, and potassium), whereas the others did not. Furthermore, a regression analysis revealed that only FACIT-F scores were independently associated with serum albumin levels in these patients, while the SF36-V scores were not. However, larger-scale studies are needed to validate and extend our conclusion.

The prevalence of fatigue determined here (up to 65%) is consistent with the data in the existing literature (between 49% and 92% in ESRD patients) (*Almutary, Bonner & Douglas, 2013*). However, the fatigue severities determined by the different instruments in this study vary significantly from those reported by others. *Bonner, Wellard & Caltabiano, 2010* found
a moderate fatigue level (mean FSS, 4.4) among patients with advanced CKD. Jhamb et al., also discovered that the mean FACIT-F and SF36-V scores among chronic dialysis patients were 17.5–34.5 and 15–50 (40.9 ± 22.5), respectively (*Jhamb et al., 2009*; *Jhamb et al., 2013*). Other researchers revealed FACIT-F and SF36-V scores of 35.7 ± 11.8 and 61.7 ± 19, respectively, in a small group of CKD patients (*MacDonald et al., 2012*). In our cohort, the mean FSS, FACIT-F, and SF36-V scores were 1.8 ± 1.2, 41.1 ± 9.9, and 77 ± 15, respectively (Table 2); the former two scales derived slightly lower fatigue severity compared to those of the other reports, whereas the latter yielded similar scores. The relatively lower fatigue severity as assessed by the FSS and FACIT-F might have resulted from the Chinese translations that potentially produce score levels better than those from the original English versions. This point underlines the importance of translation and the potential differences in utility of the common fatigue-assessing instruments for patients with ESRD.

Fatigue is a subjective symptom that can occur at any disease stage. However, a universal definition of fatigue is still lacking, which prompts the development of multiple instruments to measure it. These instruments differ in the number of items they contain, their psychometric properties, and the dimensions they assess. It has been proposed that fatigue is an uncomfortable experience that falls along the continuum of exhaustion at one end and full of energy at the other end (*Lee, Hicks & Nino-Murcia, 1991*), and the SF36-V and LFS are designed to assess these two dimensions accordingly (*Ream & Richardson, 1996*). *Lenz et al. (1997)* proposed that factors contributing to discomfort such as fatigue could be divided into physiological, social, and psychological aspects, each with complex interactions. The FQ, which attempts to cover different parts of fatigue influences, comprises mental and physical dimensions. Finally, both qualitative and quantitative approaches to fatigue are employed by FSI. Among the seven instruments used in this study, we found that the results of both unidimensional (FACIT-F) and multidimensional (SF36-V) tools exhibited fair associations with serum albumin levels and frailty severity (SFS scores) in ESRD patients. However, the FACIT-F outperformed the SF36-V in the regression analysis; consequently, the FACIT-F might be a better option for measuring fatigue in ESRD patients on chronic dialysis.

We found that the LFS and BFI identified fatigue in a high proportion of ESRD patients, but the fatigue severity measured by these two scales did not correlate significantly with the important clinical parameters of these patients; on the contrary, the FACIT-F and SF36-V identified fatigue in fewer ESRD patients, but the severity they measured exhibited strong associations. There may be several reasons for this phenomenon. First, the content of fatigue-assessing instruments might play a role in determining the scores of individual patients. Second, it is likely that multi-item and uni-dimensional measures of fatigue might outperform briefer but multi-dimensional measures in the current cohort. This phenomenon could result from our smaller sample size, and a more focused fatigue-assessing instrument is required to detect the real prevalence and severity of fatigue among these patients. Finally, there might also be differences in the quality of Chinese translation between these instruments, rendering FACIT-F a better instrument than the other comparators.

The SF36-V is the most widely used fatigue instrument in ESRD patients; however, there are concerns about its ability to capture the negative influences of fatigue, including the interference of concentration and motivation (*Unruh, Weisbord & Kimmel, 2005*). On the other hand, the FACIT-F, by measuring multiple aspects of fatigue, might uncover the true impact of fatigue among ESRD patients. The utility of the FACIT-F has been validated in patients with different illnesses such as cancer and rheumatologic disorders (*Yellen et al., 1997*; *Chandran et al., 2007*). In ESRD patients, FACIT-F scale scores have been shown to correlate significantly with the presence of cardiovascular diseases, depression, poorer exercise tolerance, and serum albumin levels (*MacDonald et al., 2012*; *Jhamb et al., 2013*). Our findings further extend the applicability of the FACIT-F in ESRD patients by demonstrating its superiority with regard to the associations with vital outcome-related parameters, including serum albumin and frailty severity in this population.

There are few options available to reduce fatigue in patients with CKD/ESRD. Evidence suggests that exercise can improve cardio-respiratory function, increase muscle strength, and potentially restore vitality in patients with CKD (*Smart et al., 2013*). Among ESRD patients, intra-dialytic exercise might help them regain physical fitness and improve QoL (*Johansen, 2007*). Consequently, exercise training can be a useful approach for ameliorating fatigue in this population.

This is the first study to compare different fatigue-assessing instruments in ESRD patients and consider their associations with clinically important parameters. Our findings can guide subsequent researchers and clinicians in choosing a questionnaire to screen for fatigue in this population. However, this study is also restricted by its modest cohort size and lack of data on patient survival. Our conclusion is applicable to the Chinese translated versions only; whether the same conclusion can be extrapolated to fatigue-assessing instruments in other languages, including the original English versions, is unclear. Thus, additional studies are needed to validate and extend the applicability of our results.

## CONCLUSION

Patients with ESRD have a disproportionately high prevalence of fatigue compared to those with other illnesses due to their high comorbidity burden and the uremic milieu accompanying renal function loss. Multiple instruments exist to evaluate fatigue severity, but comparisons among them have not been attempted until now. We found that the FACIT-F scale exhibits better correlations with important outcome-associated parameters in ESRD patients than the other six examined tools. Subsequent studies are needed to confirm our results.

## ACKNOWLEDGEMENTS

We express our gratitude to the staff members of the Second Core Lab of the Department of Medical Research of NTUH for their technical support. The members of the COhort of GEriatric Nephrology in NTUH include Dr. Chia-Ter Chao, Dr. Chih-Kang Chiang, Dr. Chung-Jen Yen, Dr. Yu-Chien Hung, Dr. Su-Hsuan Hsu, Dr. Chih-Yuan Shih, Dr.

Chun-Fu Lai, Dr. Ding-Cheng Derrick Chan, Dr. Jenq-Wen Huang, Dr. Kuan-Yu Hung, Dr. Tzong-Shinn Chu, and Dr. Sheng-Jean Huang.

### Funding
This study is financially supported by National Taiwan University Hospital (NTUH) (NO. 104-S2684 and NO. 105-N3206). The funders had no role in study design, data collection and analysis, decision to publish, or preparation of the manuscript.

### Grant Disclosures
The following grant information was disclosed by the authors:
National Taiwan University Hospital (NTUH): 104-S2684, 105-N3206.

### Competing Interests
The authors declare that there are no competing interests.

### Author Contributions
- Chia-Ter Chao and Jenq-Wen Huang conceived and designed the experiments, performed the experiments, analyzed the data, contributed reagents/materials/analysis tools, wrote the paper, prepared figures and/or tables, reviewed drafts of the paper.
- Chih-Kang Chiang performed the experiments, analyzed the data, wrote the paper, prepared figures and/or tables, reviewed drafts of the paper.

### Human Ethics
The following information was supplied relating to ethical approvals (i.e., approving body and any reference numbers):

The current study is approved by the Institutional review board of the National Taiwan University Hospital (NTUH) (NO. 201403006RINB).

### Data Availability
The raw data will be uploaded as a File S1.

### Supplemental Information
Supplemental information for this article can be found online at http://dx.doi.org/10.7717/peerj.1818#supplemental-information.

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
