# Peer review of "Functional assessment of chronic illness therapy—the fatigue scale exhibits stronger associations with clinical parameters in chronic dialysis patients compared to other fatigue-assessing instruments"

_PeerJ, doi:10.7717/peerj.1818_

## Round 0.1 · original submission · Major Revisions

The reviewers have found some merit in the manuscript but also feel that a variety of issues need to be addressed prior to the paper being considered for publication in PeerJ.

I therefore strongly urge you to take on board the comments for these reviewers prior to resubmitting the article to PeerJ.

Reviewer 1 ·

Basic reporting

The authors have completed a retrospective, cross-sectional study investigating the identification of fatigue via 7 fatigue questionnaires in patients with ESRD receiving dialysis. This is an original study which fits within the scope/mission of the journal. The manuscript is well prepared, although a number of edits are recommended.
There are however a number of weaknesses in the methodology, primarily the participants selected. The authors state “and 50% of them had diabetes”. Therefore you essentially have 2 cohorts: ESRD receiving dialysis and ESRD patients with T2dm receiving dialysis.

Fatigue is well documented in a number of chronic diseases/disorders and their respective treatments, therefore the authors should be more stringent in defining their subjects due to potential confounding effects of fatigue aside from ESKD and dialysis. Given approximately 50 percent of their participants were diabetic (assuming T2dm), the authors should also complete data analysis on 3 groups: combined, non-diabetic and diabetic to determine if there are differences. I believe this will strengthen the manuscript.

Additionally, the authors have omitted albumin and creatinine from the Introduction section, as well as published studies which are relevant (Karakan et al., Clinical Nephrology, 2011).

Experimental design

The design is appropriate for the study undertaken and represents original research. The research question is clearly defined, description of the Methods needs improvement. The study has institutional review approval ((NTUH; NO. 201403006RINB) with "verbal" consent obtained. Concerns exist with the participants who (most likely) will have co-morbidities which will effect fatigue. Sub-group analyses is highly recommended based upon approximately one half of the participants being diabetic (assumed T2dm), other co-morbidities are non-specified. Additionally, a power analysis for determination of subject numbers is missing (should be incorporated into the manuscript)

Validity of the findings

The data is robust and sound, however as stated above may have significant confounding variables. Once sub-group data analysis is completed, the authors will have more valid findings their are reporting.

The focus of the authors study was the identification of fatigue in ESRD patients undergoing dialysis however there is no mention of exercise as an intervention (for example) which has demonstrated efficacy in reducing fatigue in this cohort (Smart et al., J Sci and Med in Sport, 2013 and Johanses, J. American society of Nephrology, 2007). It would be most appropriate for the authors to conclude with intervention(s) which can help offset the fatigue induced by dialysis. Ross et al., (1989, Am J Nephrology) may also be a useful read.

Additional comments

Line 1 Suggest “had” should be replaced with “have”
Line 10 the authors discuss that regression analysis of their outcome measures was completed with serum albumin, it is recommended that the authors include what level of serum albumin was used in parentheses as an indicator of fatigue. The same should be included for creatinine (gender specific)
Line 10 “confirm” should be replaced with “investigate”
Line 11 the authors have written “on average higher age than the general dialysis population…) This is somewhat confusing as readers may not interpret as a higher average age in their hospital/clinic dialysis population or the general population receiving dialysis (overall)
Line 12 the authors have stated that 50% of their participants have clinically diagnosed diabetes, they should specify whether they are referring to T2dm or T1dm or both. This is also problematic with regards to their subject population as a poor HbA1c will contribute to fatigue (Segersted et al., 2015, J. Clinical and Translational Endocrinology).
ABSTRACT
40 Suggest rewording “not yet been attempted” --> “yet to be investigated”
45 suggest rewording “we studied” --> “we investigated”
45-46 the authors discuss they have investigated the correlation between “fatigue severity and selected clinical/biochemical parameters” it is recommended the authors specify briefly the primary outcome/biochemical parameters in parentheses
48 The authors start the ”Results” section with “The enrolees were…” It is recommended the authors begin the results section with the total number of participants in the study. For example a total of 46 ESRD patients, higher average age……”
48-49 the authors have written “on average higher age than the general dialysis population…) This is somewhat confusing as readers may not interpret as a higher average age in their hospital/clinic dialysis population or the general population receiving dialysis (overall)

The authors should specify the type of diabetes of the participants in parentheses, for example and 50% of them had T2dm diabetes.
50-51 the authors have used both general (p < 0.01) and specific p-values (p = 0.03) in the abstract and throughout the manuscript. It is highly recommended the authors be consistent.
INTRODUCTION
63 the authors state that patients with ESR D are living longer “than ever”, it is recommended the authors briefly elaborate upon this sentence. For example due to increased medical care and/or early diagnosis and/or improved treatment…

Suggest rewording ”but this extension” --> “however this increased survival”
64 the authors use the wording “cost”, are they referring to financial? It is recommended they specify.
The authors start the next sentence with the acronym “ESRD”, it is generally recognized that you do not begin any sentence with an acronym therefore recommend spelling out in these instances
65 the authors use the wording “course”, are they referring to disease progression and/or treatment? It is recommended to specify
69 suggest reword “their” --> ESRD patients
75 the authors had started the sentence with an acronym suggest spell out
87 suggest reword “but” --> however
the authors also use the terminology “the most optimal instrument “, are they referring to the most valid and/or reliable instrument? Suggest reword
General the authors have used albumin and creatinine (among other) primary outcome measures however, they have not included any information regarding these clinical/biochemical parameters in the “Introduction” section of the manuscript. It is highly recommended they include information pertaining to biochemical markers to identify fatigue.
MATERIALS/METHODS
99-100 were there any exclusion criteria to participants eligible for the study?
104 the authors state that blood samples were obtained and sent for analysis, was this completed in-house or sent to a commercial laboratory. It is recommended the authors specify
110 the authors state that trained research and assistants administered all of the fatigue assessments, was this completed in one day/session or over a number of days? Additionally was the order of administration of the surveys randomized or with a specific order? This should be specified within the manuscript
113 the authors state that “Most of these tools…” The authors should specify the number in parentheses (n = ?).
114 the authors state that “the literature with proven fair internal and external validity”, are the authors referring to “established”? It’s so they should specify the validity in parentheses
122 the authors state “… Features the advantages of brevity and avoidance of English-based idioms that tend to confuse recipients”. Perhaps include the word “Chinese”? So this reads “… Tend to confuse Chinese recipients….”
126 the authors state with fair validity, they should specify in parentheses
150 the authors have omitted a power-analysis for their selected subject numbers, this should be completed.
151 suggest reword to past tense “are” --> “were”
RESULTS
176 the authors referred to “previously reported diagnostic threshold for each questionnaire” these thresholds should be included in text and/or in Table 2
178 the authors state “did the same”, is recommended this be reworded to “identified fatigue”
188 the authors have (all p < 1.01), the word “all” is not required as the sentence states “Results from all seven…..”
DISCUSSION
212 it is recommended the authors delete the word “Here” and begin the sentence with “”We administered…..”
213 suggest reword “found that 15 – 65% had fatigue” as I believe you are summarizing that the seven questionnaires you administered found between 15 to 65% of participants were identified with fatigue?
22-221 references and details should be included with this sentence
226 the authors have used the word “group”, usually studies are not referred to as groups, recommend reword .
264 suggest re-word but --> however
276 recommend delete first sentence in this line
278 the authors have stated “Our findings can guide subsequent researchers….” It is recommended that the authors also include clinicians. For example “our findings can guide subsequent researchers and clinicians in choosing….”
The authors should reword their use of “optimal”.
279 this is the first mention that this was a ”pilot study”, recommend delete
280 suggest reword “more” --> “additional”

GENERAL the focus of the authors study was the identification of fatigue in ESRD patients undergoing dialysis however there is no mention of exercise as an intervention (for example) which has demonstrated efficacy in reducing fatigue in this cohort (Smart et al., J Sci and Med in Sport, 2013 and Johanses, J. American society of Nephrology, 2007).
REFERENCES
the authors have selected a total of 30 references which appear to be most appropriate to support their manuscript
TABLE 1 the authors have center justified and right justified the clinical features, co-morbidity ESRD origin and laboratory data, this should be left justified. Given the authors have included a number of clinical/laboratory data, globular filtration rate (GFR) should also be included as this identifies the participants stage of kidney disease. It may be beneficial for the authors to include both the stage (1, 2, 3A, 3B, 4, 5) and GFR (mls/min)
TABLE 2 the authors have center justified the parameters, this should be left justified
TABLE 3 the authors have omitted what the values represent. It is recommended the authors clarify. Additionally they authors have chosen to use general p-values (<0.01) as opposed to specific p-values. If they decide in the manuscript to use specific p-values, then this should be incorporated into Table 3 for consistency
TABLE 4 the authors once again have a mix of general and specific p-values, it is recommended consistency throughout the tables and manuscripts

·

Basic reporting

The writing is clear and appropriate for the most part. At times, English-language phrasing seemed not to flow naturally, and could benefit from copyediting by a native English-speaker. For example, in Introduction, is this really the way an English language questionnaire is worded: (“Do you feel fatigue/tiresome/that you are lacking energy?”)?

What does it mean to say that FACIT-F and SFS were shown to have "fair" validity (Wang et al, e.g.)? This makes them seem problematic. Are you sure there was not at least "good" validity? this may be important given the thrust of this article is about the superiority of the FACIT-F in particular, relative to other measures. If it is only "fair" what would that mean? I suggest you use the word "good" in these cases.

"Here the SF36-V and FACIT-F scores were reversed, indicating higher fatigue severity with increased scores" ... why must this be done. It causes confusion when people try to compare your results with official scoring. I strongly urge you to present the results with properly scored totals. It is not difficult to accommodate the reader with creative use of displays. Examples exist in the literature (see EORTC and PROMIS papers....both systems have mixed direction scoring)

Experimental design

In methods section, please specify which of the questionnaires were translated previously, and which the team had to translate. Also in discussion acknowledge that a limitation of the study is that these conclusions relate to the Chinese translations and should be evaluated in other languages, including the original English versions.

Validity of the findings

I disagree with this interpretation in the discussion: "The relatively lower fatigue severity as assessed by the FSS and FACIT-F might have resulted from the unidimensional nature of these two scales, which fails to cover the spectrum of physical impairments induced by fatigue, as well as the higher number of items they contain, which potentially desensitizes the patients to these questionnaires (Minton & Stone, 2009)."
Neither one of these statements has any basis in data. I would remove them. More likely is the possibility that the translations produced resultant score levels that were higher (better) that those of the original English versions. In addition, there is no support for the idea that the FSS and FACIT-Fatigue measure non-physical fatigue, or that multidimensional assessment provides unique information by dimension, except in those cases where a fully distinct dimension is measures (such as cognition or emotion), but these are not measures of fatigue.

The speculation on lines 252-263 is not very likely to be the case, in my opinion. The fatigue levels reported in this sample were not very high, and so suggesting that FACIT might be more responsive to higher fatigue defies logic. Similarly, the speculation about age seems to be made only because there was an age relationship with FACIT-F and SF-36V. I suggest a simpler interpretation which is that a multi-item, unidimensional measure of fatigue outperformed briefer, less-unidimensional measures because the sample was small and therefore the effect required more focused measurement to obtain it. It may also be the case that the Chinese translation of the FACIT-F was of better quality, producing more valid scores than the comparator instruments. This can be mentioned as a limitation or suggestion for future research.

Additional comments

Please report all instrument scores as they are supposed to be scored per their manual instructions. Reversing scores has led to much confusion in the literature, and it is not difficult to present these results with their proper scoring.

---

## Round 0.2 · accepted · Accept

Congratulations on successfully completing the ammendments you have made to the two reviewer's comments on the previous version of this manuscript.

Reviewer 1 ·

Basic reporting

The authors have made significant improvements to their manuscript (in very timely manner), which included subsample re-analyses of data. Furthermore, they have addressed all of my recommended/required changes and suggestions.

I therefore recommend this manuscript for publication in PeerJ. I believe this manuscript will make a significant contribution to the literature in the area of assessing fatigue in patients with ESRD undergoing hemodialysis.

Experimental design

see "Basic Reporting" section above

Validity of the findings

see "Basic Reporting" section above

Additional comments

see "Basic Reporting" section above